# Evaluation of *Listeria monocytogenes* Dissemination in a Beef Steak Tartare Production Chain

**DOI:** 10.3390/foods14193372

**Published:** 2025-09-29

**Authors:** Simone Stella, Carlo Angelo Sgoifo Rossi, Francesco Pomilio, Gabriella Centorotola, Marina Torresi, Alexandra Chiaverini, Maria Filippa Addis, Cristian Bernardi, Martina Penati, Clara Locatelli, Paolo Moroni, Silvia Grossi, Viviana Fusi, Paolo Urgesi, Erica Tirloni

**Affiliations:** 1Department of Veterinary Medicine and Animal Sciences, University of Milan, Via dell’Università 6, 26020 Lodi, Italy; simone.stella@unimi.it (S.S.); carlo.sgoifo@unimi.it (C.A.S.R.); filippa.addis@unimi.it (M.F.A.); cristian.bernardi@unimi.it (C.B.); martina.penati@unimi.it (M.P.); clara.locatelli@unimi.it (C.L.); paolo.moroni@unimi.it (P.M.); silvia.grossi@unimi.it (S.G.); viviana.fusi@unimi.it (V.F.); paolo.urgesi@unimi.it (P.U.); 2IZSAM, Istituto Zooprofilattico Sperimentale dell’Abruzzo e del Molise G. Caporale, Via Campo Boario, 64100 Teramo, Italy; f.pomilio@izs.it (F.P.); g.centorotola@izs.it (G.C.); m.torresi@izs.it (M.T.); a.chiaverini@izs.it (A.C.)

**Keywords:** steak tartare, beef carcasses, *Listeria monocytogenes*, MALDI-TOF, whole-genome sequencing, antibiotic resistance

## Abstract

This study evaluated the diffusion of *Listeria monocytogenes* (LM) in a beef steak tartare production chain, aiming to (1) evaluate *Listeria* spp. diffusion in finishing farms supplying beef cattle, (2) evaluate LM prevalence in carcasses, and (3) map LM diffusion in the production plant. A detection rate of 6/76 was observed in the farm, while carcasses after skinning and before refrigeration tested positive in 19/30 and 11/30, respectively. During tartare production, 57/154 meat and 35/191 environmental samples tested positive. A total of 114 LM isolates were characterized via a whole-genome sequencing approach. Five clonal complexes (CCs) and seven sequence types (STs) were identified, with CC9-ST580 being the most prevalent. Four clusters were identified from both the slaughtering and production phases. Genes related to resistance to fosfomycin, quinolones, sulfonamides, lincosamide, and tetracycline were detected. Two hypervirulent strains (CC6-ST6 and CC2-ST145), harboring a full-length *inlA*, several virulence genes, and stress islands, were detected. Stress Survival Islet 1 was found in almost all the isolates. The wide diffusion of LM in steak tartare requires the management of some critical phases of the production chain (mainly slaughtering); genomic methodologies could be useful in describing the circulation and virulence of LM strains.

## 1. Introduction

*Listeria monocytogenes* (LM) is a crucial pathogen in the production chain of Ready-To-Eat (RTE) foods; its importance as a foodborne pathogen is increasing due to the growing number of reported cases in recent years in Europe and its diffusion in many food categories. Among food categories, meat and products thereof showed a relatively high prevalence of this pathogen; for instance, a prevalence of 4.1% was found in RTE bovine meat [1].

Within RTE meats, steak tartare is very popular. It is obtained from raw ground meat, thus being considered a perishable foodstuff [2]. It is traditionally considered a fresh product that is prepared by the butcher for direct consumption. However, its use as a pre-packaged product has increased in recent years, often with a shelf life of 10–15 days assigned by producers. In this context, it is important to highlight both the presence and the growth potential of LM in the product. A recent study in the Netherlands [3] identified an association between the consumption of steak tartare and an increased risk of human infections. Moreover, LM isolates of cattle origin (particularly those linked to fresh meat and its production environments) were estimated to be the primary cause of human listeriosis. The synergic action of the cold chain, product packaging (often ensuring an anaerobic environment via vacuum or skin vacuum packaging), and composition is usually sufficient to prevent the pathogen from growing quickly [4,5,6]. Nevertheless, it must be noted that steak tartare is not specifically labeled as a “risk foodstuff”. Thus, all consumers could be exposed to the presence of the pathogen, including Young, Old, Pregnant, and Immunocompromised individuals (YOPI), who can sometimes develop severe invasive listeriosis even if exposed to relatively low counts [7].

LM can be introduced in the product and into the production environment through raw materials, abiotic vectors, or personnel [8]. LM circulation pathways in food production chains are highly variable; in the case of steak tartare, contamination could occur at various stages. At the farm level, cattle can be contaminated via several environmental sources. The main sources of animal infection are soil, feed, bedding, and water troughs, as well as contact with other infected wild or farm animals or transmission via arthropod vectors. The consumption of silage is a significant source of infection [9]. Cattle can function as a reservoir, with a variable prevalence of positive animals from 1.5% to more than 30% [1,10]. At the slaughterhouse, the main source of LM is animals carrying the pathogen on their skin. Carcass contamination can then occur through operators, tools, or slaughterhouse facilities [7]. Worldwide, several studies [7,9,11,12,13] have been conducted to determine the prevalence of LM in samples taken from cattle. This pathogen has been identified on cattle hides (13.3%) and carcasses (2.8%) at processing plants in the United States [11]. In a slaughterhouse investigation regarding the contamination of bovine hides, a prevalence of 9.9% was found among 1033 animals examined, with a significant difference between the prevalence rates of the two considered plants, thus showing the importance of the application of good manufacturing practices [12].

Finally, deboning, sectioning, and production operations can lead to contamination from external sources. Once the pathogen has entered a production environment, its persistence can be favored by the complexity of the equipment (e.g., mincing and mixing machines), which results in difficulty carrying out sanitation procedures, leading to the formation of long-term survival niches for LM. Even if optimal hygiene procedures are applied in production plants, some production phases of steak tartare can spread the pathogen, as many meat cuts are processed on the same production line, thus resulting in frequent contamination of the final product.

This study aimed to evaluate the dissemination of LM in a beef chain intended for steak tartare production. Very few studies are addressed in this field and consider the entire chain. The first step was to evaluate the diffusion of *Listeria* spp. in the pre-slaughter phase (by sampling farm and slaughterhouse environments, animals, and carcasses). The second step was focused on beef carcass contamination by LM during slaughtering. Finally, the third step was to map the diffusion of LM in the tartare production plant (by sampling raw matter, product, and production environment). Characterization of the isolates was performed to identify potential sources and highlight their circulation within the steak tartare production plant and the previous phases of the production chain.

## 2. Materials and Methods

### 2.1. Experimental Plan

The experimental study involved farms and a plant located in Northern Italy; three phases were sampled, namely, farming (where young bulls and beef cows were raised during the finishing period), slaughtering (at a plant receiving the animals from the selected farms), and sectioning/production of ground meat and meat preparations (in the same plant where animals were slaughtered). Sampling was performed during various sessions between 2021 and 2023. The study included three steps (Figure 1).

#### 2.1.1. Sampling—First Step: Evaluation of *Listeria* spp. and LM Diffusion in Finishing Farms and on Animals

In the first step, the diffusion of LM and other *Listeria* spp. in two beef cattle farms and on animals slaughtered at the plant intended for steak tartare production was investigated. During two sampling sessions, a total of 76 samples were taken: (a) at the farms (43 samples) from litter, feeding troughs, manure, feed (unifeed, silage, hay, core feed, and corn meal), and cattle skin, choosing animals at the end of the growing period, and (b) at the slaughterhouse (33 samples), on the day when the animals previously sampled were slaughtered. The following points were considered: hide of live animals in the pre-slaughter race, carcasses after slaughtering, and drains of the slaughtering area (Table 1).

Samples of litter, manure, and feed were collected in sterile bags. Samples from troughs, animal skin, carcasses, and drains were collected using sterile sponges soaked in 10 mL of sterile saline (NaCl/Tryptone 0.85/0.1%); when possible, 10 × 10 cm areas were sampled.

#### 2.1.2. Sampling—Second Step: Evaluation of LM Diffusion on Slaughtered Carcasses

In the second step, carcasses used for the production of steak tartare were investigated for LM: two sampling sessions were conducted, and during each session, 15 carcasses were sampled using sterile sponges on four 10 × 10 cm areas. Each carcass underwent surface sampling in two stages: at the end of the skinning phase and at the end of the slaughtering process, prior to refrigeration. This approach was employed to assess the impact of different phases of the slaughtering process on the spread of the pathogen on carcasses.

#### 2.1.3. Sampling—Third Step: Evaluation of LM Diffusion in the Production Plant

In the third step, an investigation was conducted to detect LM in the meat processing facility, aiming to detect production phases at risk for its diffusion in the product. A total of 154 samples were taken from raw matter and meat during each phase to detect the eventual point of the production chain where contamination could occur (see Table 2). Beef samples were taken before sectioning (primal cuts), after removal of the surface sheath, after dressing, during hardening, after cubing, after mixing, after forming, after the final hardening phase, and from the final product (beef steak tartare). For sample collection, sterile knives (for whole cuts) and sterile bags were used. Moreover, 191 operative environmental samples were collected using sterile sponges, when possible, on 10 × 10 cm areas from surfaces in contact with meat (Food Contact Surfaces: FCSs), transfer points, and drains (identified as Non-Food Contact Surfaces: NFCSs). In the sectioning area, tables and knives were chosen as FCSs, while in the production area, the FCSs sampled were knives; tables; tubs; racks; machinery used for “peeling”, cubing, mixing, and forming; and small equipment (see Table 3). Transfer points included rack wheels (in sectioning and production areas), tub wheels, handles of racks and doors, and turnstiles. Drains were sampled in the meat sectioning, hamburger production, and steak tartare production areas. Ten pre-operational samples were also taken to evaluate the role of the surfaces as contamination sources: samples were taken from tables, cubing, mixing, and forming machines and tubs.

### 2.2. Sample Analyses

All the samples were immediately refrigerated upon collection and transported to the laboratory; the analyses were performed on the same day.

Sponges were supplemented with 90 mL of Half Fraser broth (Scharlab, Sentmenat, Spain), while 25 g of the other samples were taken from the sterile bags, put in sterile stomacher bags with a filter strip, and supplemented with 225 mL of Half Fraser broth. All the samples were homogenized for 60 s in a Stomacher 400 (Seward Medical, London, UK). Half Fraser broth was incubated at 37 °C for 24 h. Then, 100-microliter streaks were plated on Rapid’L.mono agar plates (Bio-Rad, Hercules, CA, USA) for LM detection, following the AFNOR BRD 07/04-09/98 method [14], and on Palcam agar plates (Scharlab) for the detection of *Listeria* spp. (only in the first experimental step).

For identification purposes, from the plates obtained during the first sampling step, up to ten typical *Listeria* spp. colonies from each sample were picked and subcultured on Palcam agar plates. In the second and third sampling steps, one suspect LM colony for each sample was picked and subcultured on Rapid’L.mono agar plates.

### 2.3. Isolates Identification

A total of 363 isolates were obtained from 165 LM- and/or *Listeria* spp.-positive samples (241 presumptive *Listeria* spp. from 43 samples in the first step, 30 presumptive LM from 30 samples in the second step, and 92 presumptive LM from 92 samples in the third step) and were submitted for species identification via matrix-assisted laser desorption/ionization time-of-flight mass spectrometry (MALDI-TOF MS) with a Bruker Biotyper System (MBT, Bruker Daltonics GmbH, Bremen, Germany) in duplicate. For identification, a small portion of an isolated colony was transferred onto duplicate wells on the surface of an MBT target plate with a sterile toothpick, overlaid with 1 μL of 70% formic acid, let dry, overlaid with 1 μL of α-cyano-4-hydroxycinnamic acid (HCCA) solution (50% acetonitrile, 47.5% water, 2.5% trifluoroacetic acid), and let dry. MBT target plates were then processed for mass spectra acquisition with a Microflex LT/SH mass spectrometer (Bruker Daltonik GmbH) in the positive mode. Each target plate also included one well with the Bacterial Test Standard for instrument calibration (Bruker Daltonik GmbH, Bremen, Germany). The obtained spectra were interpreted against the MBT Compass^®^ Library Revision H (2023), equipped with the MBT HT subtyping module for the confirmation and identification of LM and discrimination from other *Listeria* spp. [15]. The MBT has been certified according to the Official Method of Analysis program (OMA) of AOAC International. The species identification method is also certified according to the ISO 16140-part 6 standard, Certificate N2017LR75 [16].

### 2.4. Whole-Genome Sequencing of LM Isolates

The 63 isolates identified as LM from steps 1 and 51 selected LM isolates from step 3 (including isolates from each typology of meat or environmental sample), resulting in a total of 114 isolates, were stored at −20 °C in Microbank (PRO-LAB diagnostics, Biolife Italiana S.r.l., Milan, Italy) and sent to the Italian National Reference Laboratory for *L. monocytogenes* to be further characterized using a whole-genome sequencing (WGS) approach. The DNA extraction was performed using the QIAamp DNA Mini Kit (QIAGEN, Hilden, Germany), according to the manufacturer’s instructions; after the lysis phase, lysozyme from chicken egg solution was added (20 mg/mL) (Sigma-Aldrich, Milan, Italy). The DNA was then quantified using the Qubit dsDNA HS (High Sensitivity) Assay Kit (Invitrogen, Carlsbad, CA, USA), while the quality parameters (A260/A230 and A260/280 ratios) were assessed via Biospectrometer fluorescence (Eppendorf, Milan, Italy). WGS was performed as previously reported by Centorotola et al. [17], using the Illumina platform (Illumina, San Diego, CA, USA). WGS data analysis of the LM sequences obtained was performed using an accredited in-house pipeline (https://github.com/genpat-it/ngsmanager/ (accessed on 8 May 2025)), and quality was assessed according to ISO 23418:2022 [18].

The multilocus sequence typing (MLST) and the core genome MLST (cgMLST) were performed according to Pasteur’s reference schemes (https://bigsdb.pasteur.fr/, accessed on 8 May 2025).

Moreover, all the LM genomes were characterized in silico using BIGSdb-*Lm* database tools (https://bigsdb.pasteur.fr/, accessed on 8 May 2025) for “Virulence”, “Antibiotic resistance”, “Metal and disinfectant resistance”, and “Stress islands” queries.

According to the cgMLST results, the threshold of ≤7 allelic distance was considered for the cluster definition, as reported by Moura et al. [19]. Furthermore, the cgMLST results were visualized using the Interactive Tree of Life (iTOL) (https://itol.embl.de/, accessed on 16 May 2025), grouping virulence, stress, and antibiotic resistance factors across all the 114 LM isolates. The software GrapeTree (https://achtman-lab.github.io/GrapeTree/MSTree_holder.html (accessed on 16 May 2025)) was used to obtain the Minimum Spanning Tree (MSTreeV2) [20] for the CC9-ST580 isolates. The LM genomes were deposited in DDBJ/ENA/GenBank under the BioProject PRJNA1254504, for the *Lm*62 isolate, and PRJNA1252953 for the other 113 LM isolates.

### 2.5. Antimicrobial Susceptibility of LM Isolates from Farm and Production Plant

The 63 LM isolates obtained from step 1 analyses were tested for antimicrobial susceptibility using the disk diffusion method described by the European Committee on Antimicrobial Susceptibility Testing (EUCAST) [21]. The panel of 17 antibiotics included the following: amoxicillin–clavulanic acid (30 µg; AMC), ampicillin (10 µg; AMP), cephalothin (30 µg; KF), ceftriaxone (30 µg; CRO), ciprofloxacin (5 µg; CIP), erythromycin (15 µg; E), gentamicin (10 µg; CN), linezolid (30 µg; LZD), meropenem (10 µg; MEM), nalidixic acid (30 µg; NA), penicillin G (10 µg; P), piperacillin (30 µg; PRL), rifampin (30 mg; RD), trimethoprim–sulfamethoxazole (1.5 µg–23.5 µg; SXT), tetracycline (30 µg; TE), tigecycline (15 µg; TGC), and vancomycin (30 µg; VA). The isolates were classified as susceptible, intermediate, or resistant to each antibiotic according to the breakpoints proposed by the EUCAST and the Clinical and Laboratory Standards Institute [22] for LM, where available, or for *Staphylococci* spp., *Escherichia coli* ATCC 25922, and *Staphylococcus aureus* ATCC 25923, which were used as quality control strains.

### 2.6. Statistical Analyses

The data obtained in the first and third steps of the study were analyzed via a chi-square test or exact Fisher’s test to evaluate the differences in *Listeria* spp. and LM detection rates in the different sample typologies; the significance threshold was set at *p* = 0.05.

## 3. Results

### 3.1. First Step: Evaluation of Listeria spp. and LM Diffusion in Finishing Farms and on Animals

In the first phase of the study, the diffusion of LM and other *Listeria* spp. in cattle farms and on animals slaughtered at the same plant was investigated. The results of the analyses performed on the samples taken from the farms are shown in Table 1. A low detection rate was evidenced for LM (4.4% of the total number of samples), which was isolated only from the litter samples in farm A. The evaluation for the presence of *Listeria* spp. in the farm environment showed a wide diffusion (41.8%) in the samples taken from cattle skin, the environment, and feed. A higher detection rate was observed in farm B (47.8%) compared with farm A (35%), although this difference was not statistically significant (*p* = 0.589). The samplings performed at the slaughterhouse were planned in order to find the same groups of animals that were present in the finishing farms when the previous evaluation was performed. *Listeria* spp. were frequently detected in the samples (75.8%); half of the samples were taken from the skin of live animals, and 76.9% of the carcasses harbored *Listeria* spp., and as a consequence, they were constantly isolated from the drains of the slaughtering area. LM contamination was evidenced in 12.1% of the samples. Although the pathogen was not detected on live animals, its presence on the carcasses at the end of the slaughtering process was relatively frequent (15%). As expected, it was also isolated from the drains (20%).

### 3.2. Second Step: Evaluation of LM Diffusion on Slaughtered Carcasses

In the slaughtering phase, a total of 30 carcasses were tested after the skinning phase and at the end of the slaughtering line, before refrigeration. LM was detected in 23 out of 30 carcasses (76.7%), without any clear difference between the sampling sessions (12/15 and 11/15, respectively), thus suggesting a likely constant introduction of the pathogen into the plant by contaminated animals and the possibility of frequent contamination of the raw matter used for the production of steak tartare. A higher prevalence was detected in carcasses after the skinning phase (19/30, 63.3%) compared with the end of the slaughtering process (11/30, 36.7%), although this difference was not statistically significant (*p* = 0.071).

The results obtained from the single 30 carcasses (Figure 2) showed that 7 were LM positive at both sampling steps; 12 carcasses that tested positive after the skinning phase became negative at the end of the process, whereas 4/30 became positive only at the end of the process. The other 7 carcasses were LM negative at both sampling steps. These data could suggest a potential role of slaughtering operations (particularly carcass rinsing and dressing for the removal of dirty parts), although no intervention can completely eliminate the presence of the pathogen on carcass surfaces.

### 3.3. Third Step: Evaluation of LM Diffusion in the Production Plant

A total of 345 samples (154 meat samples and 191 environmental samples) were analyzed to detect LM in the third experimental step. The results are shown in Table 2 (meat samples) and Table 3 (environmental samples).

The results of the analyses performed on meat samples during the production of steak tartare showed a high LM detection rate (37%); when analyzing the trend of the contamination along the production line, a high prevalence of the pathogen was already evidenced in the raw matter used (primal cuts; 39%). In the first stages of the production process, the detection rate decreased (22%, 17%, 6%, and 10% after sheath removal, dressing, hardening, and cubing, respectively), presumably due to the removal of more contaminated surface parts. A further evident increase was observed in mixed products (87.5%) and remained at high levels in the last part of the process (50% and 61% after forming and hardening, respectively), resulting in frequent contamination of the final product (72%).

The data obtained from the environmental samplings showed a wide circulation of the pathogen on equipment and operative surfaces in all the production areas of the plant during the production of steak tartare. A global prevalence of 18.3% (35 out of 191 samples) was detected when considering all the samples. Similar detection rates (around 16.7–16.8%) were detected in both sectioning (7/42) and tartare production (24/143) areas. The pathogen was mainly isolated from FCS (21.2%, 29/137), with a higher rate (also if not significantly, *p* = 0.158) compared with NFCS (11.1%, 6/54); this trend was evidenced also considering the sectioning area (21.9%, 7/32 vs. 0/10) and the production area (21.0%, 22/105 vs. 5.3%, 2/38) separately. Among the samples, drains were considered as they can be used as sentinel points for the detection of pathogens within production plants: in this case, LM was detected only in the hamburger production area, with a very high rate (4/6). As for meat samples, it was possible to follow the trend of the contamination along the production chain: a relatively low contamination rate was detected in the first phases (2/9 on the peeling machine, 2/18 on dressing equipment, 1/9 racks where the cuts are hung, and 1/11 on the cubing machine), whereas a very high value was observed on the mixing machine (8/9) and forming machine (4/9). The comparison of the meat and environmental sample data showed a parallel trend among the two typologies of samples (Figure 3).

The results of the preoperative samplings, performed on clean surfaces before the beginning of the production, did not show the presence of LM on the FCS tested (tables, cubing, mixing, and forming machines and tubs; 0/10 samples).

### 3.4. Listeria Isolates Identification

A total of 363 isolates from 165 samples were submitted for identification via MALDI-TOF. From the samples taken during the first step of the study, a total of 241 isolates were analyzed: 231 isolates belonged to the genus *Listeria*, whereas 10 isolates were not identified as *Listeria* spp. The species distribution is shown in Table 4. A significant difference in the species identified among the isolate population belonging to farm and slaughterhouse samples was observed (*p* < 0.001). *L. innocua* was the most abundant species in both farm and slaughterhouse samples, but a marked difference was evidenced between the two phases (84.6% vs. 42.8% of the isolates, respectively). Among the samples from the farms, *L. innocua* was the only species isolated apart from litter samples, where LM was also detected. A higher diffusion of LM and *L. welshimeri* was detected in the samples taken at the slaughterhouse and, in particular, in the drains. This difference between the two stages could be due to various environmental sources acting at the slaughterhouse, and to the slaughtering of several animals from different farms on the same day. The identification of presumptive LM isolates obtained from the second and third sampling steps (30 and 92 isolates, respectively) was confirmed via MALDI-TOF MS.

### 3.5. Characterization of LM Isolates

A total of 114 LM isolates were obtained and characterized. All the information on positive samples and related LM isolates is reported in Appendix A.

Among all the LM isolates, the MLST analysis identified five different clonal complexes (CCs) and seven different sequence types (STs): CC9-ST580 (95/114), CC70-ST70 (10/114), CC9-ST9 (5/114), CC9-ST3286 (1/114), CC6-ST6 (1/114), CC121-ST121 (1/114), and CC2-ST145 (1/114). The most abundant clone detected was CC9-ST580 (95/114), observed in various types of samples, including environmental swabs, carcasses, meat, and steak tartare, collected from different sampling sites during different sampling sessions. The CC9-ST9 clone was found only in one sample of animal skin, from sampling session 1, in the slaughterhouse area. The new ST assigned by Pasteur, ST3286, was observed in one of the five isolates from the same positive sample (carcasses) from sampling session 2. Referring to CC6-ST6 and CC121-ST121 strains, these clones were found in drains, from two different sites sampled during sampling session 2, while the CC2-ST145 strain was detected in a meat-cubed sample from the steak tartare production area.

The cgMLST results and the annotation of the main virulence, antibiotic, and stress resistance factors found in the 114 LM isolates are reported in Figure 4. In detail, CC6-ST6 carried the highest number of virulence genes (67/93), followed by CC9 (61/93), CC2 (58/93), CC121 (58/93), and CC70 (53/93). All the 114 isolates had complete *L. monocytogenes* Pathogenicity Island 1 (LIPI-1), but only the CC6-ST6 isolate (*Lm*62) showed the presence of the additional and complete LIPI-3. A full-length *inlA* was confirmed for CC70-ST70, CC6-ST6, and CC2-ST145 isolates. On the contrary, for all the CC9 and CC121-ST121 isolates, an internal stop codon in *InlA* was detected, and a full-length *inlB* was observed in all the 114 LM isolates.

Referring to stress resistances, all the CC70 and CC9 isolates carried complete Stress Survival Islet 1 (SSI-1), while SSI-2 was observed in the CC121-ST121 isolate only, together with the presence of different *Tn6188* transposons, such as *Tn6188_qac, Tn6188_tetR, Tn6188_tnpA, Tn6188_tnpB,* and *Tn6188_tnpC.*

Finally, with regard to antimicrobial resistance factors, all the isolates tested carried intrinsic core genes, particularly *fosX*, *norB*, *sul,* and *lin*. Moreover, the additional *tetM* and *tetS* genes were observed in CC9-ST9 isolates only.

According to the cgMLST results, four clusters were observed among the 114 LM isolates tested. All the related information is reported in Appendix A and Figure 4.

In detail, one cluster was referred to CC70-ST70 isolates (cluster I), isolated from two different litter samples, in the same farm. Moreover, two clusters were confirmed for CC9-ST580 isolates from different samples and sampling sites. The first CC9-ST580 cluster (cluster II) was observed among 19 LM isolates obtained from carcasses and drains, sampled in the slaughterhouse area, but also from environments and meat related to the steak tartare production area. The second CC9-ST580 cluster (cluster III) involved 74 isolates obtained from the carcasses and drains sampled in the slaughterhouse from the dicing machine and drains sampled in the hamburger production area, but also in steak tartare production environments and the steak tartare final product. All the details related to CC9-ST580 clusters and isolation sources are reported in Figure 5. Finally, a CC9-ST9 cluster (cluster IV) was found, related to LM isolates from the same sample (animal hide).

### 3.6. Antibiotic Susceptibility of LM Isolates

All the 63 LM isolates from the step 3 analyses were submitted for genomic characterization (see Appendix A, isolates N. 1 to 63) and antibiotic susceptibility testing (Table 5). Among β-lactams, a general low resistance rate to penicillins (amoxicillin, ampicillin, penicillin, and piperacillin) and cephalotin was observed, whereas about 1/5 of the isolates showed resistance to meropenem, and a very high resistance rate for ceftriaxone was detected. Among quinolones, rare resistance was detected for ciprofloxacin, whereas nalidixic acid was always ineffective. Rifampin, linezolid, and vancomycin were the only molecules that inhibited all the LM isolates; considering the other antibiotic classes, a low prevalence of resistance was observed for tigecycline, and an increasing rate was detected for gentamicin, tetracycline, erythromycin, and trimethoprim–sulfamethoxazole, with the last one showing a very high diffusion of resistance.

## 4. Discussion

This study focused on the contamination pathways of steak tartare production by LM, considering the importance of this pathogen for RTE foods, the wide diffusion of steak tartare among consumers, and the absence of lethal or partially lethal treatments during the production process [3,7]. Our data showed a high prevalence of the pathogen within the product (more than 1/3 of the samples analyzed). A variable prevalence was detected in previous studies on similar foodstuffs, with both lower and higher (up to 55%) values [1,23,24]. The frequent contamination by LM can originate from different sources, owing to its ubiquitous nature and its environmental persistence. In the case of steak tartare production, a direct link could be inferred between cattle entering the slaughterhouse, the carcasses, the sectioned meat used for production, and the final product.

In the first step of this study, the potential source of LM entering the slaughterhouse was investigated. The prevalence found at the two farms considered was low and limited to the litter. Thus, no primary source of contamination was found. Some studies showed the role of silage as a source of LM for the animals and that it can be spread through feces [9]. In this study, no LM-positive samples were detected in any of the feed samples. *Listeria* spp. were found to be widely present in the environment, feed, and animals, indicating the possible circulation of these ubiquitous bacteria. The results obtained from the farm environment were reflected by the data obtained from animal hide (sampled at both the farms and the slaughterhouse), showing the diffusion of *Listeria* spp. but without detecting LM. Nevertheless, the carcasses at the end of the slaughtering process showed a moderate prevalence of the pathogen (about 15%). This result, in parallel with the detection of LM from the drains of the slaughtering area, could be influenced by the concomitant slaughtering of many other animals coming from various farms. A marked difference within the *Listeria* spp. population was evidenced between the different environments, with a very high prevalence of *L. innocua* in the farms and an increased prevalence of LM, associated with the presence of *L. welshimeri* at the slaughterhouse; this pattern was observed in both sampling sessions.

When focusing on the slaughtering phase, a wide diffusion of LM on cattle carcasses slaughtered at the plant was evident, in agreement with the variable contamination rates found by different authors, in some cases with very high values [11,12,13,25]. Starting from the hypothesis of a frequent presence of the pathogen on the hide of cattle, the role of good manufacturing practices during slaughtering was evident. Our data showed the critical role of the skinning phase, as more than 60% of the skinned carcasses were LM positive, whereas this rate decreased to about 1/3 at the end of the process, although some carcasses became contaminated only at the end of slaughtering. It has to be noted that the procedure applied at the plant considered in this study included a dressing and washing phase (using potable water), which was applied at the end of the process (before sampling) to remove little blood clots or bone fragments. The utility of carcass washing is debated, as it could act as both a positive (removal of gross contamination) and negative intervention (difficulty in detecting gross contamination) [26]; in our study, it was mainly beneficial, as it could be useful for carcasses with a high contamination rate after skinning.

In the last phases of the steak tartare production chain (third step of the study), the role of the meat used in the production process was evaluated. The results showed the high prevalence of the pathogen in the sectioned meat (almost 40%). As the contamination of meat cuts is limited to their surface, the first stages of the production process were characterized by a low prevalence of LM, thanks to the dressing operated to improve the quality of steak tartare. Nevertheless, the final part of the production line inevitably represents a spreading phase, owing to the bottleneck represented by machines used for many meat cuts, leading to a diffusion of contamination among a high number of final product units. The role of meat cuts as contamination sources for the final product was also highlighted by the results of environmental samplings: the presence of the pathogen was indeed detected mainly on FCS, and the trend of LM detection on the surfaces nearly mimics the one observed on the meats. Moreover, positive samples were only found during production. Even if the number of samples was low, the data suggest the absence of a meaningful role of bacterial persistence within the plant, whereas a frequent entry with meats can be hypothesized, with the diffusion in different areas of the same plant (e.g., sectioning area, steak tartare production, and hamburger production).

The LM isolate population obtained from the samplings was characterized via WGS, showing the presence of different clones; in particular, CC9 was the most abundant. As reported in the literature, LM CC9 and CC121 are strongly associated with food and the meat processing environment [27,28], but not with clinical cases, demonstrating their hypovirulence [29]. Among the CC9 strains, CC9-ST580 was the main clone found in this study, reported in the literature from different sources, such as meat and meat products [30], pork meat [31], and Polish artisanal cheeses [32]. LM CC9 and CC121 are known in the literature as hypovirulent clones, characterized by *InlA* gene truncations, the major feature linked to the reduction in the virulence [29]. However, despite their reduced virulence potential, CC9 and CC121 have the increased ability to resist and persist in food production areas, even for years, under environmental stresses, thanks to several resistance factors [27,28,33]. In a study by Guidi et al. [28], two different persisting CC9 clusters, persisting for 4 and 2 years, were reported in a pork meat processing plant located in Central Italy. Furthermore, the presence of SSIs confers stress resistance to LM strains; in particular, SSI-1, frequently observed in strains belonging to different clones [33], including CC9, is linked to environmental stress resistance, such as low pH, high osmolarity, bile, and nisin [34]. On the other hand, SSI-2, found in the CC121 isolate only, is known to be linked to alkaline and oxidative stress tolerance [35]. Moreover, CC121 harbors different *Tn6188* transposons, such as *Tn6188_qac* for tolerance to benzalkonium chloride, a quaternary ammonium compound known to be effective against LM, frequently used in food processing plants [36].

CC70-ST70 was the second most common clone found in this study, but limited data are available in the literature for this clone. A study conducted by Dreyer et al. [37] reported LM CC70-ST70 both in small ruminant feces and in their farm environment.

Among the singletons observed in this study, LM CC6-ST6 was isolated from environmental samples. This clone was reported in the literature from chicken skin samples [38] and also from Polish artisanal cheeses studied by Pyz-Łukasik et al. [32], in which CC6-ST6 was the dominant one found. Together with LM CC1, CC2, and CC4, CC6 strains are known as hypervirulent and strongly associated with clinical infection cases [27]. The increased virulence profile of this clone is linked to the *Listeria* Pathogenicity Island 3 (LIPI-3) presence, aside from LIPI-1, ubiquitous in all the LM strains. While LIPI-1 is known to be involved in the intracellular infection cycle of LM, LIPI-3 encodes listeriolysin S, an additional hemolysin that is a hemolytic and cytolytic factor, increasing the potential virulence of the strain [39,40]. While a full-length *inlB* was observed in all the LM isolates analyzed in this study, a full-length *inlA* was reported only in CC70-ST70, CC2-ST145, and CC6-ST6 isolates. The *inlA* and *inlB* genes, encoding for Internalin A and Internalin B, respectively, are known to be crucial for the invasion of host cells [41].

The CC2-ST145 isolated in this study from meat cubes sampled in the steak tartare production was reported in the literature, mainly related to meat and meat products [30,42]. Moreover, Lachtara et al. [43] recovered ST145 from all the sources involved in their study, such as raw meat and RTE food, but also in food production environments.

Considering the virulence of LM isolates considered in this study, the abundant presence of hypovirulent clones, such as CC9, carrying a complete SSI-1, could help LM strains to adapt to stress and survive and persist for years in the food production environment. As reported in this study, for the CC9-ST580 isolates, two different clusters were observed, involving different types of samples, such as the environment and the steak tartare final product, and different sampling sites, including both the slaughterhouse and hamburger and steak tartare production areas. The identification of CC9-ST580 clusters may suggest the presence of a transmission pathway, along the entire food production chain, or indicate contamination events occurring at different production and processing levels, possibly involving environmental or animal sources. This aspect should be taken into account in order to identify the possible clone hotspots within the establishments considered in this study and evaluate the diffusion of cc9-st580 strains both in the plant and in food products, also evaluating the presence of persistent strains. On the other hand, hypervirulent clones, such as CC6 and CC2, in the slaughterhouse environment (drains) and the steak tartare production area (meat-cubed), respectively, pose a risk for consumers, especially linked to the possible contamination of RTE meat products, including steak tartare.

Regarding the genetic antibiotic resistance factors observed in the LM isolated in this study, the presence of core genes was confirmed in all the isolates, particularly the *fosX*, *norB*, *sul*, and *lin* genes related to resistance to fosfomycin, quinolones, sulfonamides, and lincosamide, respectively [44,45]. Resistance to quinolones (nalidixic acid) and sulfonamides was also detected via the Kirby–Bauer method in all the CC-STs tested. Trimethoprim–sulfamethoxazole is a potential second-choice treatment for LM infection, but a very high resistance rate was shown: this phenomenon has been linked to plasmid-mediated mechanisms by Charpentier and Courvalin [46]. Additional resistance genes, such as *tetM* and *tetS*, were observed in this study for CC9-ST9 isolates. These acquired genes are associated with resistance to tetracyclines (other potential second-line drugs), particularly the *tetM* gene [45]. The phenotypic test confirmed tetracycline resistance in the same five CC9-ST9 isolates, but it was also detected in four other isolates belonging to CC9-ST580. In a study conducted by Moura et al. [45], a higher prevalence of these acquired antimicrobial resistance genes was observed in food isolates, but contrasting results were obtained by previous authors [11,47,48,49]. The occurrence of multi-drug-resistant LM isolates was detected in different CC-STs: two isolates (one CC9-ST580 and the only C6-ST6 isolate), both isolated from the drains in the slaughterhouse, showed resistance to nine antibiotics, and other CC9-ST580 and CC9-ST9 isolates, taken from hides and carcasses during slaughtering, were resistant to seven antibiotics. A low resistance rate to penicillins (amoxicillin, ampicillin, penicillin, and piperacillin) was observed; this result was in agreement with previous studies by Wieczorek et al. and Conter et al. [11,47]. Resistance to these antibiotics has already been observed and should be considered with care, as they are considered first-line drugs for the treatment of listeriosis [46,49]. Considering the other β-lactams, a relatively high resistance was detected for meropenem (a potential second-choice treatment), as already observed by Thønnings et al. [50]. They also highlighted the low efficacy of cephalosporines: in our study, first-generation cephalosporin (cephalotin) showed good efficacy, but more than 4/5 LM isolates were resistant to a third-generation one (ceftriaxone) due to the different spectrum of these molecules. Most of the results obtained for the other antimicrobials were expected and confirmed by previous results, e.g., the efficacy of linezolid and the rare resistance to ciprofloxacin; for other antimicrobials, such as vancomycin and rifampin (which were effective against all the isolates) and erythromycin and gentamicin (which had a relatively high resistance rate), variable results were obtained by other authors [11,47,49]. In particular, resistance to gentamicin, observed in 10% of LM isolates, is of concern, as this molecule is considered among the first-choice molecules for listeriosis treatment [46]. The overall antimicrobial resistance profile obtained indicated a higher prevalence of resistance (including multi-drug resistance) among LM isolates compared with previous results from LM strains isolated from food and food processing environments, which is considered a potential concern [11,47,51].

Future research should focus on continuous monitoring and genomic characterization of LM isolates to identify persistent or hypervirulent clones and support targeted mitigation strategies to reduce consumer risk. It has to be considered that the risk posed by LM is dose dependent, with a “safe” threshold of 100 CFU/g, set by EC Regulation 2073/2005 [52]. Nevertheless, the risk level can vary greatly based on consumer sensitivity, with the potential exposure of YOPI people. In this light, the role of Food Business Operators is extremely important, and is being enhanced by the incoming legal amendments [53], aiming to improve the safety of foodstuffs for their whole shelf life. Thus, a synergy of various hurdles (such as packaging and growth of biopreservative like lactic acid bacteria) can be positively applied by producers to prevent significant risks for consumers.

## Figures and Tables

**Figure 1 foods-14-03372-f001:**
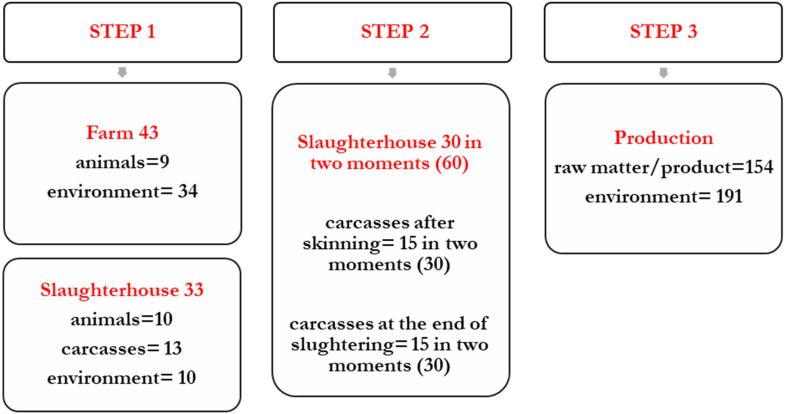
Sampling scheme.

**Figure 2 foods-14-03372-f002:**
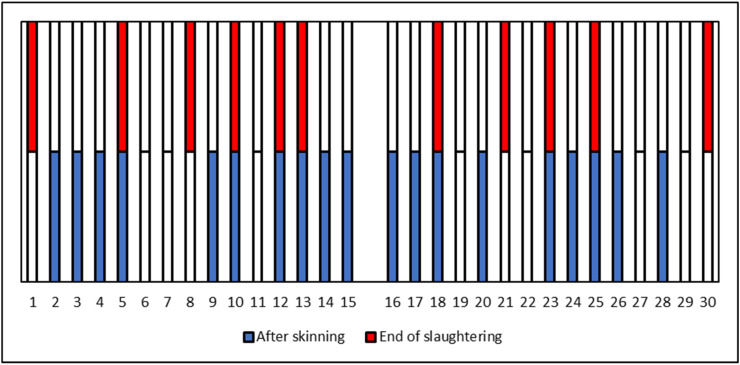
Detection of LM on carcasses after skinning and at the end of the slaughtering process.

**Figure 3 foods-14-03372-f003:**
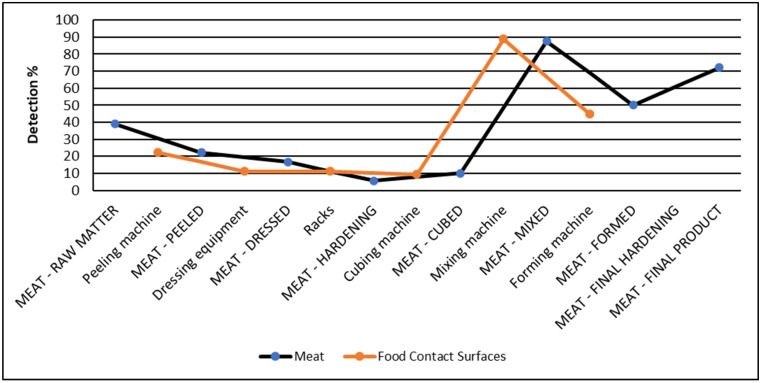
Trend of the detection rates of LM on meat (capital letters) and environmental samples during steak tartare production phases.

**Figure 4 foods-14-03372-f004:**
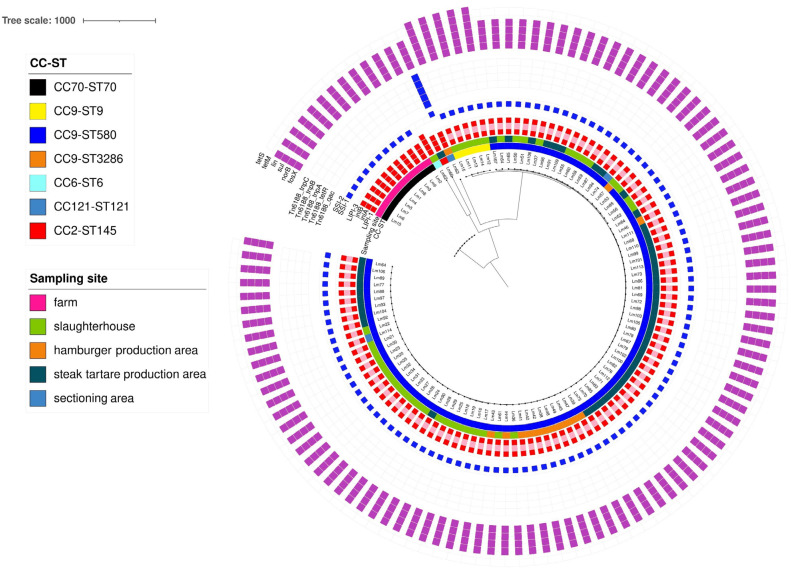
Core genome multilocus sequence typing (cgMLST) tree grouping the major virulence and resistance factors across the 114 LM isolates. Virulence (red square), antibiotic resistance (purple square), and stress resistance (blue square) genes are shown in the heatmap. Colored square: presence of the gene; light red: presence of a premature stop codon and truncated *inlA*. The visualization of the gene profiles and gene presence/absence according to their cgMLST was visualized using the Interactive Tree of Life (iTOL) (https://itol.embl.de/, accessed on 15 May 2025).

**Figure 5 foods-14-03372-f005:**
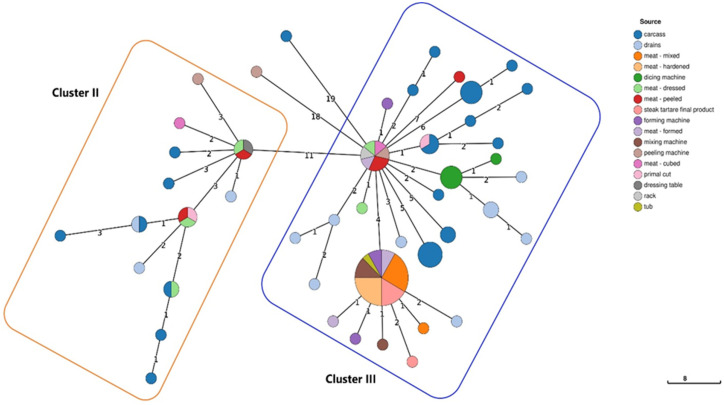
Minimum Spanning Tree (MST) based on the cgMLST profiles of LM CC9-ST580, colored according to the isolation source. The CC9-ST580 clusters are highlighted with boxes: the yellow box is for cluster II, while the blue box is for cluster III. The MST was visualized using GrapeTree (https://github.com/achtman-lab/GrapeTree, accessed on 15 May 2025).

**Table 1 foods-14-03372-t001:** Detection of *Listeria* spp. and LM in samples taken from farm and slaughterhouse.

Location	Sample	*Listeria* spp.	LM
Farm	Cattle skin	2/9	0/9
	Litter	3/9	2/9
	Feeding trough	4/5	0/5
	Manure	0/5	0/5
	Feed	9/15	0/15
	Total	18/43	2/43
Slaughterhouse	Live cattle skin	5/10	0/10
	Carcasses	10/13	2/13
	Drains	10/10	2/10
	Total	25/33	4/33
Global		43/76	6/76

**Table 2 foods-14-03372-t002:** Detection rate of LM in meat samples taken from the steak tartare production line.

Sampling Phase	Detection Rate (No. of Positive/Total Samples)
Raw matter (primal cuts)	7/18
After sheath removal	4/18
After dressing	3/18
During hardening	1/18
After cubing	2/20
After mixing	7/8
After forming	9/18
After final hardening	11/18
Final product	13/18
Total	57/154

**Table 3 foods-14-03372-t003:** Detection rate of LM in environmental samples during sectioning and steak tartare production phases.

Area	Surface Type	Sample	Detection Rate (No. of Positive/Total Samples)
Sectioning	FCS	Sectioning tables	3/24
		Racks	4/8
	NFCS	Racks wheels	0/8
		Drains	0/2
Hamburger production	NFCS	Drains	4/6
Steak tartare production	FCS	Peeling machine	2/9
		Dressing tables	1/9
		Dressing knives	1/9
		Racks	1/9
		Cubing machine	1/11
		Mixing machine	8/9
		Tubs	2/9
		Forming machine	4/9
		Scale	2/9
		Small equipment	0/22
	NFCS	Rack wheels	1/9
		Tub wheels	1/9
		Rack handles	0/4
		Door handles	0/8
		Turnstiles	0/4
		Drains	0/4
Total FCS			29/137
Total NFCS			6/54
Global			35/191

FCS: Food Contact Surface; NFCS: Non-Food Contact Surface.

**Table 4 foods-14-03372-t004:** *Listeria* species identification in samples from farm and slaughterhouse.

Phase	Sample	Species		
		*L. innocua*	*L. welshimeri*	*L. monocytogenes*
Farm	Animal skin	5 (100%)	0	0
	Litter	5 (33.3%)	0	10 (66.7%)
	Feeding trough	14 (100%)	0	0
	Feed	31 (100%)	0	0
	Total	55 (84.6%)	0	10 (15.4%)
Slaughterhouse	Animal skin	6 (50.0%)	0	6 (50.0%)
	Carcasses	20 (39.2%)	0	31 (60.8%)
	Drains	45 (43.7%)	45 (43.7%)	13 (12.6%)
	Total	71 (42.8%)	46 (27.7%)	49 (29.5%)
Global		126 (54.5%)	46 (19.9%)	59 (25.5%)

**Table 5 foods-14-03372-t005:** Antibiotic susceptibility of LM isolates.

Antibiotic	S	I	R
Nalidixic acid (NA)	-	-	63 (100%)
Amoxicillin–clavulanic acid (AVG)	59 (93.7%)	1 (1.6%)	3 (4.8%)
Ampicillin (AMP)	62 (98.4%)	1 (1.6%)	-
Cephalothin (KF)	60 (95.2%)	1 (1.6%)	2 (3.2%)
Ceftriaxone (CRO)	9 (14.3%)	-	54 (85.7%)
Ciprofloxacin (CIP)	61 (96.8%)	2 (3.2%)	-
Erythromycin (E)	54 (85.7%)	-	9 (14.3)
Gentamicin (CN)	57 (90.5%)	1 (1.6%)	5 (7.9%)
Linezolid (LZD)	63 (100%)	-	-
Meropenem (MEM)	51 (81.0%)	-	12 (19.0%)
Penicillin (P)	62 (98.4%)	-	1 (1.6%)
Piperacillin (PRL)	61 (96.8%)	-	2 (3.2%)
Rifampin (RD)	63 (100%)	-	-
Trimethoprim–sulfamethoxazole (SXT)	36 (57.1%)	-	27 (42.9%)
Tetracycline (TE)	54 (85.7%)	4 (6.4%)	5 (7.9%)
Tigecycline (TGC)	61 (96.8%)	-	2 (3.2%)
Vancomycin (VA)	630 (100%)	-	-

S = sensitive, I = intermediate, and R = resistant.

## Data Availability

The datasets presented in this study can be found in online repositories. The names of the repository/repositories and accession number(s) can be found in the article/Appendix A and at https://www.ncbi.nlm.nih.gov/genbank/ (accessed on 25 September 2025) under the BioProjects PRJNA1254504 and PRJNA1252953.

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
