# Peer review of "Evaluation of Listeria monocytogenes Dissemination in a Beef Steak Tartare Production Chain"

_foods, 2025, doi:10.3390/foods14193372_

Round 1
Reviewer 1 Report
Comments and Suggestions for Authors
The work evaluated the presence of L. monocytogenes along the prodution chain of steak tartare, all the way from the farms to the final product. The work raises the concern with this type of product that does not have any killing step to eliminate this and other pathogens.
There are a few issues that authors could adress to improve the presentation:
1 - line 26: "quite all isolates" sounds strange. Rephrase
2 - line 49: "The combination of respect of cold chain" should be rephrased
3 - line 238: state the % to be consistent with the rest of the presentation
4 - Table 1: what is the difference between these carcasses and those from the Second step? A diagram of the whole sampling process would be helpful.
5 - line 252-3: clarify. Before you said 30 out of 60.
6 - Line 257-60: a fluxogram of the process and the sampling points is needed so we can understand what is happening with each carcass
7 - line 321: Maldi-TOF results usually come with a confidence (score) to it. Please share the data.
Additional points:
1 - have the authors perfomed any additional confirmation to the Maldi-Tof results? It would be interesting to have a second confirmation of the isolates
2 - include a figure with the sampling scheme following the production chain.
3 - consider including a figure showing the industry and the positivity of each site within the production plant. It would be interesting to see all the positives considering the layout of the food factory.
4 - consider discussing the risk of consuming this kind of product, particularly for the at risk population. A discussion on the legal microbiological limits of L. monocytogenes RTE foods is also expected, in light of the results obtained in this study.
Author Response
1 - line 26: "quite all isolates" sounds strange. Rephrase this was done
2 - line 49: "The combination of respect of cold chain" should be rephrased this was done
3 - line 238: state the % to be consistent with the rest of the presentation all the paragraph was implemented and percentages clarified
4 - Table 1: what is the difference between these carcasses and those from the Second step? A diagram of the whole sampling process would be helpful. this was included
5 - line 252-3: clarify. Before you said 30 out of 60. this was clarified
6 - Line 257-60: a fluxogram of the process and the sampling points is needed so we can understand what is happening with each carcass this was done
7 - line 321: Maldi-TOF results usually come with a confidence (score) to it. Please share the data. Confidence for all identifications was >2, if needed we can add a supplementary table
Additional points:
1 - have the authors perfomed any additional confirmation to the Maldi-Tof results? It would be interesting to have a second confirmation of the isolates analysis was done in duplicate, moreover part of these strains were confirmed with WGS
2 - include a figure with the sampling scheme following the production chain. this was done
3 - consider including a figure showing the industry and the positivity of each site within the production plant. It would be interesting to see all the positives considering the layout of the food factory. due to privacy policy of the factory involved it was not possible to include
4 - consider discussing the risk of consuming this kind of product, particularly for the at risk population. A discussion on the legal microbiological limits of L. monocytogenes RTE foods is also expected, in light of the results obtained in this study. this was done

Reviewer 2 Report
Comments and Suggestions for Authors
The manuscript “Evaluation of Listeria monocytogenes Dissemination in a Beef Steak Tartare Production Chain” (foods-3886913) evaluated the dissemination of LM in a beef chain from pre-the slaughter phase to the tartare production plant. Then the whole genome sequencing and antimicrobial susceptibility were evaluated. The design is logical, and the novelty is sound. The language is also in good manner. Therefore, I suggest minor revision before acceptance.
Abstract: Which process was the most contamination source of Listeria monocytogenes? The results should be stated in the abstract.
The purpose and the results should be stated separately.
Introduction: The shortage of the current research can be stated.
Line 81-89, The purpose and the main content of this study should be covered.
According to the study and survey, which process was the most critical for the contamination of Listeria spp.?
L175, Why only LM from step 1 and step 3 were tested the whole genome sequencing? The reason should be stated.
L207, Why only LM from step 1 were tested the antimicrobial susceptibility? The reason should be stated.
Table 5, what are S, I, and R?
Author Response
Abstract: Which process was the most contamination source of Listeria monocytogenes? The results should be stated in the abstract. this was done
The purpose and the results should be stated separately. this was done
Introduction: The shortage of the current research can be stated. this was done
Line 81-89, The purpose and the main content of this study should be covered.this was done
According to the study and survey, which process was the most critical for the contamination of Listeria spp.? this was included
L175, Why only LM from step 1 and step 3 were tested the whole genome sequencing? The reason should be stated.
For WGS, due to cost of technique we decided to submitt to analyse only a part of strains, and in particular those strictly related to final product.
L207, Why only LM from step 1 were tested the antimicrobial susceptibility? The reason should be stated.
For AMR, we decided to submit to analyse only a part of strains and in particular step 1 because in this step we could focus on strinas related to the diffusion throught the whole chain from farm environment to slaughterhouse.
Table 5, what are S, I, and R? this was included
